# Combining diaries and accelerometers to explain change in physical activity during a lifestyle intervention for adults with pre-diabetes: A PREVIEW sub-study

Leon Klos[1,2], Gareth Stratton[1], Kelly A. Mackintosh[1], Melitta A. McNarry[1], Mikael Fogelholm[3], Mathijs Drummen[4], Ian Macdonald[5], J. Alfredo Martinez[6,7,8,9], Santiago Navas-Carretero[6,7,8], Teodora Handjieva-Darlenska[10], Georgi Bogdanov[10], Nicholas Gant[11], Sally D. Poppitt[12], Marta P. Silvestre[12,13], Jennie Brand-Miller[14], Roslyn Muirhead[14], Wolfgang Schlicht[15], Maija Huttunen-Lenz[16], Shannon Brodie[14], Elli Jalo[3], Margriet Westerterp-Plantenga[4], Tanja Adam[4], Pia Siig Vestentoft[17], Heikki Tikkanen[18], Jonas S. Quist[19,20,21], Anne Raben[17,19], Nils Swindell[1]*

1 Applied Sports, Technology, Exercise and Medicine (A-STEM) Research Centre, Faculty of Science and Engineering, Swansea University, Swansea, Wales, United Kingdom, 2 Institute of Sports and Sports Science, Karlsruhe Institute of Technology, Karlsruhe, Germany, 3 Department of Food and Nutrition, University of Helsinki, Helsinki, Finland, 4 Department of Nutrition and Movement Sciences, Maastricht University, Maastricht, Netherlands, 5 School of Life Sciences, Queen's Medical Centre, University of Nottingham, Nottingham, United Kingdom, 6 Center for Nutrition Research, University of Navarra, Pamplona, Spain, 7 CIBEROBN, Instituto de Salud Carlos III, Madrid, Spain, 8 IdiSNA, Navarra Institute for Health Research, Pamplona, Spain, 9 Program for Precision Nutrition, IMDEA Food Institute, Madrid, Spain, 10 Department of Pharmacology and Toxicology, Medical University of Sofia, Sofia, Bulgaria, 11 Department of Exercise Sciences, University of Auckland, Auckland, New Zealand, 12 Human Nutrition Unit, School of Biological Sciences, University of Auckland, Auckland, New Zealand, 13 Centro de Investigaçao em Tecnologias e Serciços de Saûde (CINTESIS), NOVA Medical School, NOVA University of Lisbon, Lisbon, Portugal, 14 School of Life and Environmental Biosciences and Charles Perkins Centre, University of Sydney, Sydney, New South Wales, Australia, 15 Department of Exercise and Health Sciences, University of Stuttgart, Stuttgart, Germany, 16 Institute of Nursing, University of Education, Schwäbisch Gmünd, Germany, 17 Department of Nutrition, Exercise and Sports, University of Copenhagen, Copenhagen, Denmark, 18 Faculty of Health Sciences School of Medicine, University of Eastern Finland, Kuopio, Finland, 19 Department for Clinical and Translational Research, Copenhagen University Hospital—Steno Diabetes Center Copenhagen, Herlev, Denmark, 20 Department of Biomedical Sciences, University of Copenhagen, Copenhagen, Denmark, 21 School of Psychology, University of Leeds, Leeds, United Kingdom

* n.j.swindell@swansea.ac.uk

**Data Availability Statement:** Ethical approval for this study was granted provided that participants' data was only accessible by the research team.

## Abstract

Self-report and device-based measures of physical activity (PA) both have unique strengths and limitations; combining these measures should provide complementary and comprehensive insights to PA behaviours. Therefore, we aim to 1) identify PA clusters and clusters of change in PA based on self-reported daily activities and 2) assess differences in device-based PA between clusters in a lifestyle intervention, the PREVIEW diabetes prevention study. In total, 232 participants with overweight and prediabetes (147 women; 55.9 ± 9.5yrs; BMI $\geq$25 kg·m$^{-2}$; impaired fasting glucose and/or impaired glucose tolerance) were clustered using a partitioning around medoids algorithm based on self-reported daily activities before a lifestyle intervention and their changes after 6 and 12 months. Device-assessed PA levels (PAL), sedentary time (SED), light PA (LPA), and moderate-to-vigorous PA (MVPA)

Participants of this study did not consent to having their data publicly available. Requests to access the data may be directed to the Swansea University College of Engineering Research Ethics Committee coeresearchethics@swansea.ac.uk.

**Funding:** The PREVIEW study received grants from the EU 7th Framework Programme (FP7-KBBE-2012), grant no: 312057; the New Zealand Health Research Council, grant no. 14/191; and the NHMRC-EU Collaborative Grant, Australia.This work was supported by a fellowship of the German Academic Exchange Service (DAAD; recipient: Leon Klos). The funders had no role in study design, data collection and analysis, decision to publish, or preparation of the manuscript.

**Competing interests:** Ian MacDonald: Principle investigator of the Nottingham site, conception of the study, editing of the manuscript. International Life Sciences Institute (ILSI) Europe: Member of Dietary Carbohydrates Task Force, Member of expert group on 'Efficacy Markers of Diabetes Risk'. Travel and subsistence paid but no attendance fees. Nature Publishing Group (Springer Nature) - Editor International Journal of Obesity. Travel and accommodation re-imbursed and honorarium paid (Amount received per annum over £5,000: yes). Nestle Research - Scientific Advisory Board. Travel and accommodation reimbursed, and honorarium paid (Amount received per annum over £5,000: yes) Mars Incorporated-Waltham Centre for Pet Nutrition Peer-review of pet nutrition research projects honorarium received (Amount received per annum over £5,000: no); Mars UK/Europe - Member of Nutrition Advisory Board, and Health and Wellbeing Committee travel and subsistence costs reimbursed. Honorarium paid to the University of Nottingham. Mars Incorporated - Member of Mars Scientific Advisory Committee. Honorarium paid to the University of Nottingham. Novozymes - Member of Science Advisory Board. Travel and subsistence costs reimbursed. Honorarium paid to the University of Nottingham. Public Health England - Member of Scientific Advisory Committee on Nutrition. Travel and accommodation re-imbursed and honorarium paid (Amount received per annum over £5,000: no) Jennie Brand-Miller: Principle investigator of the Sydney site and participated in editing of the manuscript. Jennie Brand-Miller is President and Director of the Glycemic Index Foundation and oversees a glycemic index testing service at the University of Sydney. She receives royalties for books about carbohydrates, obesity and diabetes. This does not alter our adherence to PLOS ONE policies on sharing data and materials.

were assessed using ActiSleep+ accelerometers and compared between clusters using (multivariate) analyses of covariance. At baseline, the self-reported "walking and house-work" cluster had significantly higher PAL, MVPA and LPA, and less SED than the "inactive" cluster. LPA was higher only among the "cycling" cluster. There was no difference in the device-based measures between the "social-sports" and "inactive" clusters. Looking at the changes after 6 months, the "increased walking" cluster showed the greatest increase in PAL while the "increased cycling" cluster accumulated the highest amount of LPA. The "increased housework" and "increased supervised sports" reported least favourable changes in device-based PA. After 12 months, there was only minor change in activities between the "increased walking and cycling", "no change" and "increased supervised sports" clusters, with no significant differences in device-based measures. Combining self-report and device-based measures provides better insights into the behaviours that change during an intervention. Walking and cycling may be suitable activities to increase PA in adults with prediabetes.

## Introduction

The health benefits of physical activity (PA) are well documented [1]. While research has predominantly focussed on the benefits of moderate-to-vigorous physical activity (MVPA), more recent research has shown that light physical activity (LPA) is also associated with lower morbidity and mortality [2, 3], whilst sedentary time (SED) has negative associations with health outcomes [4, 5]. Unfortunately, the measurement of PA in free-living conditions remains a challenge, not least because PA is a complex, multi-dimensional construct: as described by Piggin [6], "physical activity involves people moving, acting and performing within culturally specific spaces and contexts, and influenced by a unique array of interests, emotions, ideas, instructions and relationships". This definition highlights that PA is not only defined by its duration, intensity or associated energy expenditure but is also characterised by its pattern, type, domain (e.g., occupational, transport, leisure) and by intra-day variations, in addition to values and social factors [7]. Currently, there is no "gold standard" for measuring the spectrum of movement behaviours, which encompasses SED, LPA and MVPA, as no method can accurately encompass all aspects of PA behaviour [8, 9]. Nonetheless, device-based and self-reported PA measures are most commonly used to assess PA in free-living conditions [10].

Accelerometers are the most widely used device-based assessment of PA, especially when assessing the relationship between movement behaviours and health measures [10], showing less variability and susceptibility to bias compared to self-report measures [9, 11]. However, accelerometry assessments cannot provide insight into the context of PA and, although they capture most movement during locomotion, they do not capture some activities, most notably cycling or swimming [12].

Self-reported PA, often assessed using questionnaires or PA diaries, provides quantitative information, such as duration, intensity, and frequency of PA, as well as qualitative information about the type (e.g., walking, dancing) and the domains (e.g., occupational, leisure) of activities. Nonetheless, self-reported PA is associated with the risk of recall bias, and specifically prone to the over/under reporting of PA and sedentariness, respectively, due to social desirability, especially in intervention studies [13].

Typically, comparisons between self-reported and device-assessed PA yield only low-to-moderate correlations, primarily due to these methodological approaches exploring different constructs of PA [14, 15]. Arguably, combining these approaches in a complementary manner may therefore provide a more comprehensive insight to PA behaviours and their quantities beyond a laboratory-based setting [8, 9]. Indeed, accelerometers can detect clinically relevant changes in PA but lack information regarding which behaviours are responsible for these changes, whereas PA diaries lack precision when quantifying PA but provide context. Consequently, recent guidelines have called for a combination of self-reported and device-based PA measures [13], particularly when evaluating interventions [16].

Self-reported and device-based measures of PA have been combined in cross-sectional studies to identify PA intensities of different PA domains (e.g. occupational, recreational, household activities, active transport) [15] but no intervention study has utilised this approach to comprehensively investigate PA behaviour change and maintenance. Cluster analysis can identify subgroups of participants with distinct physical activity profiles. Identifying the underlying behaviours which lead to changes in physical (in)activity will provide valuable information for future interventions. Furthermore, a comparison of patterns can identify subgroups which use different strategies to improve their PA and may enable more tailored approaches to promoting PA in the future.

Therefore, the aims of this sub-study were i) to identify PA clusters and clusters of change in PA over the course of a lifestyle intervention based on self-reported daily activities and ii) assess differences in device-based PA between clusters.

## Methods

### Participants and setting

The "PREVention of diabetes through lifestyle Intervention and population studies in Europe and around the World" (PREVIEW) study was a multi-centre randomised trial conducted in eight countries (Denmark, Finland, Netherlands, United Kingdom, Spain, Bulgaria, Australia, New Zealand; clinicaltrials.gov NCT01777893). The PREVIEW study was approved by local Human Ethics Committees at all study sites and was conducted according to the Declaration of Helsinki (59th WMA General Assembly, Seoul, Korea, October 2008), and the ICH-GCP, The International Conference on Harmonisation (ICH) for Good Clinical Practice. All participants provided written informed consent prior to commencing screening procedures in clinic. A detailed protocol and the main results have been published elsewhere [17, 18].

However, in brief, participants were recruited across the eight study sites through direct contact with primary and occupational health care providers and through a range of media advertisements. Interested individuals were pre-screened for eligibility using the Finnish Diabetes Risk Score [19], followed by a laboratory screening. In total, 2,326 participants met the inclusion criteria of being aged 25–70 years and having both a BMI $> 25$ kg·m$^{-2}$ and pre-diabetes. Prediabetes was defined in accord with American Diabetes Association criteria [20] as either increased fasting glucose (IFG) with a venous plasma glucose concentration of 5.6–6.9 mmol·l$^{-1}$ and/or an impaired glucose tolerance (IGT) with a venous plasma glucose concentration of 7.8–11.0 mmol·l$^{-1}$ at two hours and fasting plasma glucose $< 7.0$ mmol·l$^{-1}$. The three-year randomised intervention trial consisted of an eight-week weight-loss phase during which participants were given a total meal replacement plan. During this phase, $\geq 8\%$ of initial body weight loss was required to enter the subsequent 148-week weight maintenance phase, which comprised a 2x2 factorial design with two diet and two PA arms. Participants were randomly allocated a moderate-protein (15%) medium glycemic index diet or a high-protein (25%) low-

glycemic index diet and a moderate intensity or high intensity exercise [17]. Data from three time-points during the first 12 months of PREVIEW were used in this analysis.

The PREVIEW physical activity intervention consisted of 17 group sessions over three years that were aligned to the behaviour change objectives outlined by PREMIT (PREVIEW Behaviour Modification Intervention Toolbox) [21]. Based on the trans-theoretical approach, PREMIT followed four distinct stages ((1) preliminary, (2) preparation, (3) action and (4) maintenance stages) designed generate sustained changes in physical active behaviour as opposed to prescribing exercise. The PREMIT was delivered in groups of 8–12 participants and the frequency of group sessions gradually declined from 10 visits in first six months (preparation and action phase) to three in the six months that followed (beginning of the 2.5-y maintenance phase).

## Variables

Participants wore an accelerometer and completed a PA diary seven days prior to clinical investigation days at baseline, 6- and 12-months. Furthermore, height was measured at the screening visit and body mass was measured at the clinical investigation days while participants were barefoot, wearing only light clothing and in a fasted state.

**Accelerometery.** An ActiSleep+ (ActiGraph LLC, Pensacola, FL) accelerometer was worn on the right hip, 24-hours a day, for seven consecutive days. Participants were instructed to only remove the accelerometer for water-based activities. Data were processed using 60-second epochs and non-wear was classified as 60-minutes of consecutive zeros with the allowance of interruptions for up to two minutes. After removing nocturnal sleep episodes [22], participants were required to have at least 10 hours of wear-time on four days, including one weekend day. PAL were calculated based on accelerometer counts calculated as: PAL = 0.0005882 $counts \cdot min^{-1}$ + 1.45 [23]. Time ($min \cdot day^{-1}$) spent at different intensities was categorised into SED ($<100$ $counts \cdot min^{-1}$), LPA (100–2,019 $counts \cdot min^{-1}$) and MVPA ($\geq 2,020$ $counts \cdot min^{-1}$) using Troiano cut-points [24].

**Diary data.** Participants were invited to fill out a diary for seven consecutive days during week 0, and after 6, 12, 24, and 36 months. The diary consisted of pre-selected activities, such as walking, cycling, winter sports, gymnastics, dancing, fishing and hunting, jogging, team sports, water sports, housework, gardening and occupational activities. For each activity, participants completed the duration on each day. Further, sitting time at work and at home was recorded. A minimum of 300 minutes of daily recording were required for inclusion in the analysis. Examples of the diary can be found in the supplemental S1 and S2 Figs.

## Data treatment

For each participant and time-point, only data from days with both valid accelerometer data and valid diary data were used. At least four overlapping days of valid diary and accelerometer data were required for inclusion in the analyses. Both diary and accelerometer data were calculated as mean minutes per day, at each time-point. PA change variables were subsequently calculated between baseline and six months as well as for baseline and 12 months for all diary activities and accelerometer-assessed measures.

Due to the infrequency of some activities, leisure-time activities were combined into unsupervised and supervised sports activities. Research has shown participants with obesity to have a greater propensity towards supervised activities than participants without obesity, possibly owing to regular structured sessions and extrinsic motivation and support offered by trainers and team members. Yet, self-consciousness and stigmatization have been identified as important barriers in people with obesity which may result in a preference for unsupervised activities

[25]. To account for potential preferences, unsupervised sports activity (jogging, swimming, sailing, winter sports, fishing, and hunting) and supervised sports activity (team sports, dancing, and gymnastics) were calculated separately. Both walking and cycling were analysed separately given that these were reported frequently and could not be unambiguously categorised as leisure or transport behaviours. Sitting at work and leisure sitting time were combined in a single sitting variable. Following manual inspection of the data, two outliers were removed due to implausible diary entries.

## Statistical analysis

All analyses were conducted using R. To identify PA clusters and clusters of change in PA behaviours, three separate analyses using the Partitioning Around Medoids algorithm [26] were conducted. Z-scores, which were calculated separately for each above-mentioned activity category, were used as cluster inputs. Clusters were calculated with baseline data and change from baseline to six- and 12-month data. Before each analysis, clustering tendency and the optimal number of clusters was determined. Specifically, the Hopkins statistic was calculated to determine the clustering tendency of the data. A figure above 0.5 and close to 1 indicates clustered data [27]. The average silhouette measure was used to determine the optimal number of clusters for the diary data [28]. Clusters were named after the activity (excluding sitting) with the highest z-score values.

To assess differences in PAL between baseline clusters, an ANCOVA was conducted controlling for age, sex, and BMI. A MANCOVA using Pillai's trace test statistic was calculated for the accelerometer measures (SED, LPA, and MVPA) to explore differences between clusters while controlling for age, gender, and BMI. For the six- and 12-month change clusters, differences in the magnitude of change of accelerometer-based PA measures were similarly assessed but with the addition of controlling for SED, LPA, and MVPA at baseline.

MANCOVAs with a significant cluster main effect were followed up using an ANCOVA for SED, LPA, or MVPA as appropriate for which the significance level was Bonferroni-adjusted to $\alpha = 0.0167$ to account for multiple testing. $\eta^2$ effect sizes were calculated for all ANCOVAs, with $\eta^2 > 0.001$, $> 0.06$, and $> 0.14$ indicating a small, moderate and large effect, respectively [29]. Subsequent Tukey *post-hoc* tests were conducted to ascertain significant pairwise differences between clusters ($\alpha < 0.0167$).

## Results

Of the 2,326 participants eligible for the study, 232 participants (147 women; 55.9 ± 9.5 years) had complete data for all three time-points (0, 6 and 12 months). Participants included in the analyses were, on average, approximately 4 years older and had a lower BMI and higher MVPA than the excluded participants (p<0,001) (Table 1). Further, sample-point distributions significantly differed. Specifically, 42% of included participants originated from the Finland sample, whereas no participants from Bulgaria or Australia were included (see Table 1). A detailed comparison between included and excluded participants is presented in S1 Table in the supplementary material.

## Baseline clusters

The Hopkins statistic (*H* = 0.88) indicated clustered baseline data and four distinct clusters were identified. Specifically, the first cluster "Cycling cluster" was defined by a high volume of cycling, coupled with the highest amount of occupational PA and unsupervised sports. The second cluster "Walking and housework cluster" was characterised by high volume of walking and housework and low sitting, while the third cluster "Inactive cluster" comprised individuals

**Table 1. Comparison of included and excluded participants in the longitudinal PA diary sample.**

| | Excluded participants of the PREVIEW sample (n = 1,991) | Longitudinal PA diary sample (n = 232) | p value |
|---|---|---|---|
| *Age at baseline, years* | 51.8 (11.7) | 55.9 (9.5) | < 0.001[1] |
| *Sex* | | | 0.144[2] |
| Female | 1356 (68.1%) | 147 (63.4%) | |
| Male | 635 (31.9%) | 85 (36.6%) | |
| *BMI at baseline* | 35.7 (6.7) | 32.8 (4.5) | < 0.001[1] |
| *Country* | | | < 0.001[2] |
| Denmark | 311 (15.6%) | 42 (18.1%) | |
| Finland | 179 (9.0%) | 97 (41.8%) | |
| Netherlands | 179 (9.0%) | 24 (10.3%) | |
| UK | 256 (12.9%) | 8 (3.4%) | |
| Spain | 242 (12.2%) | 46 (19.8%) | |
| Bulgaria | 364 (18.3%) | 0 | |
| Australia | 169 (8.5%) | 0 | |
| New Zealand | 291 (14.6%) | 15 (6.5%) | |

[1] Linear model ANOVA

[2] Pearson's Chi-squared test. Mean and standard deviation are reported for age and BMI.

with low activity levels and high time spent sitting. The fourth cluster "Supervised sports cluster" included individuals who largely accumulated their PA through supervised sports. PA patterns of the clusters are shown in Table 2; further characteristics of the clusters are presented in S2–S4 Tables in the supplementary material.

There was a statistically significant difference in PAL between clusters ($F(3, 225) = 8.589$, p <0.001, $\eta^2 = 0.095$), with the "walking and housework" cluster having higher a PAL than the "inactive" cluster (Fig 1).

The baseline MANCOVA also revealed differences between clusters in accelerometer-assessed SED, LPA, and MVPA ($F(9, 675) = 5.236$, p < 0.001). The follow-up ANOVA found moderate differences in MVPA between clusters ($F(3, 225) = 8.051$, p < 0.001, $\eta^2 = 0.089$). Tukey *post-hoc* analysis indicated that the "walking and housework" cluster had significantly higher MVPA than the "inactive" cluster. LPA differed between the clusters ($F(3, 225) = 6.632$,

**Table 2. Baseline clusters of time spent in self-reported physical activity behaviours in minutes per day.**

| | Inactive cluster (n = 106) | Cycling cluster (n = 23) | Walking and housework cluster (n = 61) | Supervised sports cluster (n = 42) | Total (n = 232) | Difference between clusters (p-value)[1] |
|---|---|---|---|---|---|---|
| Walking | 26.4 (20.4) | 25.1 (19.4) | 74.7 (59.9) | 34.1 (29.6) | 40.4 (41.7) | < 0.001 |
| Cycling | 1.8 (4.4) | 41.60 (23.1) | 0.8 (3.6) | 4.7 (7.0) | 6.0 (14.6) | < 0.001 |
| Unsupervised sports | 3.8 (12.9) | 11.7 (32.3) | 2.3 (7.4) | 4.1 (12.3) | 4.3 (14.9) | 0.077 |
| Supervised sports | 2.0 (3.7) | 4.9 (9.3) | 2.6 (5.3) | 22.3 (9.5) | 6.1 (9.8) | < 0.001 |
| Housework | 22.4 (28.6) | 36.1 (48.4) | 117.8 (92.6) | 60.0 (55.6) | 55.7 (70.3) | < 0.001 |
| Occupational PA | 13.4 (45.0) | 54.1 (118.5) | 13.5 (37.5) | 19.4 (68.2) | 18.5 (60.0) | 0.024 |
| Gardening | 6.8 (16.1) | 6.4 (14.3) | 11.5 (35.6) | 6.7 (16.0) | 8.0 (22.7) | 0.578 |
| Sitting | 575.8 (137.2) | 461.6 (173.9) | 338.5 (102.3) | 467.9 (134.9) | 482.6 (164.1) | < 0.001 |

All measures reported in mean (SD) min•day⁻¹. Colours of the cells based on z-transformed values used as cluster inputs (positive–green, 0 –white, negative–red, higher saturation indicates a higher absolute z-value). PA: physical activity.

[1] p-values of linear model ANOVAs between clusters.

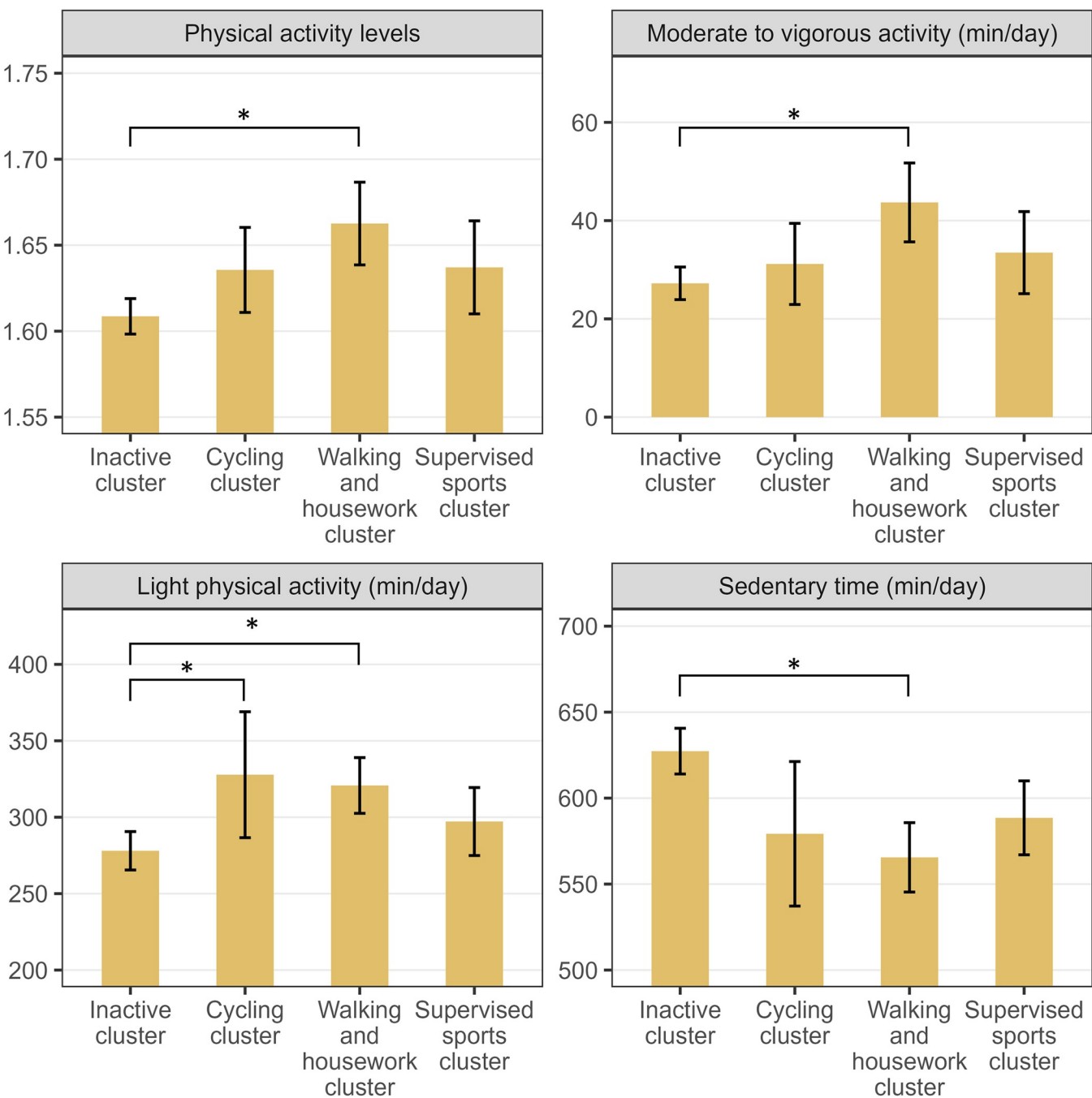

**Fig 1. Differences in accelerometer-measured physical activity and sedentary time between the baseline clusters.** Data shown as mean ± confidence intervals. Brackets with an asterisk (*) indicate significant differences between clusters.

p < 0.001, $\eta^2$ = 0.076), with the "inactive" cluster having significantly less LPA than the "walking and housework" cluster and the "cycling" cluster. Finally, there were significant differences in SED between clusters ($F$(3, 225) = 9.802, p < 0.001, $\eta^2$ = 0.108), with significantly higher SED for the "inactive" cluster compared to the "walking and housework" cluster (see also supplemental material, S5 Table).

**Table 3. Clusters of how time spent in self-reported physical activity behaviours (minutes per day) changed from baseline to six months.**

|  | Increased walking cluster (n = 73) | Increased supervised sports cluster (n = 87) | Increased cycling cluster (n = 29) | Increased housework cluster (n = 43) | Total (n = 232) | Difference between clusters (p-value)[1] |
|---|---|---|---|---|---|---|
| Walking | 23.3 (37.8) | -8.0 (33.7) | 11.6 (46.3) | -28.7 (54.2) | 0.5 (45.0) | < 0.001 |
| Cycling | 1.6 (6.7) | -2.1(10.7) | 36.9(19.2) | -1.5 (22.5) | 4.1 (18.7) | < 0.001 |
| Unsupervised sports | 3.8 (13.1) | -2.1 (23.2) | 0.3 (5.0) | 2.2 (10.7) | 0.8 (16.9) | 0.157 |
| Supervised sports | -1.9 (12.0) | 11.5 (25.6) | 2.4 (8.8) | -4.2 (10.9) | 3.2 (19.1) | < 0.001 |
| Housework | -2.6 (52.5) | -22.2 (40.5) | -21.3 (76.2) | 80.6 (85.6) | 3.1 (70.4) | < 0.001 |
| Occupational PA | 18.5 (67.1) | -1.4 (27.9) | 14.5 (55.1) | -14.5 (71.9) | 4.4 (56.2) | 0.009 |
| Gardening | 10.3 (30.8) | -5.2 (23.4) | 8.7 (35.5) | 2.8 (54.2) | 2.9 (35.1) | 0.033 |
| Sitting | -186.2 (125.3) | 30.3 (100.2) | -80.2 (97.7) | -54.9 (134.5) | -67.4 (145.5) | < 0.001 |

All measures reported in mean (SD) min•day$^{-1}$. Colours of the cells based on z-transformed values used as cluster inputs (positive–green, 0 –white, negative–red, higher saturation indicates a higher absolute z-value). PA: physical activity.

[1] p-values of linear model ANOVAs between clusters

## Change clusters baseline to six months

The Hopkins statistic ($H$ = 0.84) indicated clustered change data and four different activity change clusters were found. The "increased walking" cluster not only increased walking but also occupational PA and gardening while reducing sitting time (see Table 3). In the "increased supervised sports" cluster, total PA decreased but supervised sports increased. The "increased cycling" cluster was characterised by increased cycling, whereas the "increased housework" cluster reflected an increase in housework in exchange for reduced walking and occupational PA; further characteristics of the clusters are presented in S6–S8 Tables in the supplementary material.

There was a statistically significant difference in PAL between the six-month change clusters ($F$(3, 224) = 6.602, p <0.001, η² = 0.070), with the "increased walking" cluster having higher improvements in PAL than the "increased supervised sports" cluster (Fig 2). The six-month change MANCOVA also revealed differences in accelerometer-assessed SED, LPA, and MVPA ($F$(9, 666) = 4.274, p < 0.001). The MVPA change over six months differed between clusters ($F$(3, 222) = 4.616, p = 0.004, η² = 0.045), with significant differences between the "increased walking" and "increased housework" clusters. Differences were found for LPA ($F$(3, 222) = 7.904, p < 0.001, η² = 0.080), with the "increased walking" and "increased cycling" clusters showing a significantly greater increase in LPA than the "increased supervised sports" cluster. Differences were also found for SED ($F$(3, 222) = 3.606, p = 0.014, η² = 0.036), with the "increased walking" cluster having a higher decline in SED than the "increased supervised sports" cluster (see also supplemental material, S9 Table).

## Change cluster baseline to 12 months

The Hopkins statistic ($H$ = 0.86) indicated clustered change data and identified three different activity-change clusters. In the "increased walking and cycling" cluster sitting time was substantially reduced while in the "no change" cluster there was no change in the duration of physical activity while sitting time increased by 43 minutes (Table 4). The "increased supervised sports" cluster reduced their walking and unsupervised sports in favour of supervised sports; further characteristics of the clusters are presented in S10–S12 Tables in the supplementary material.

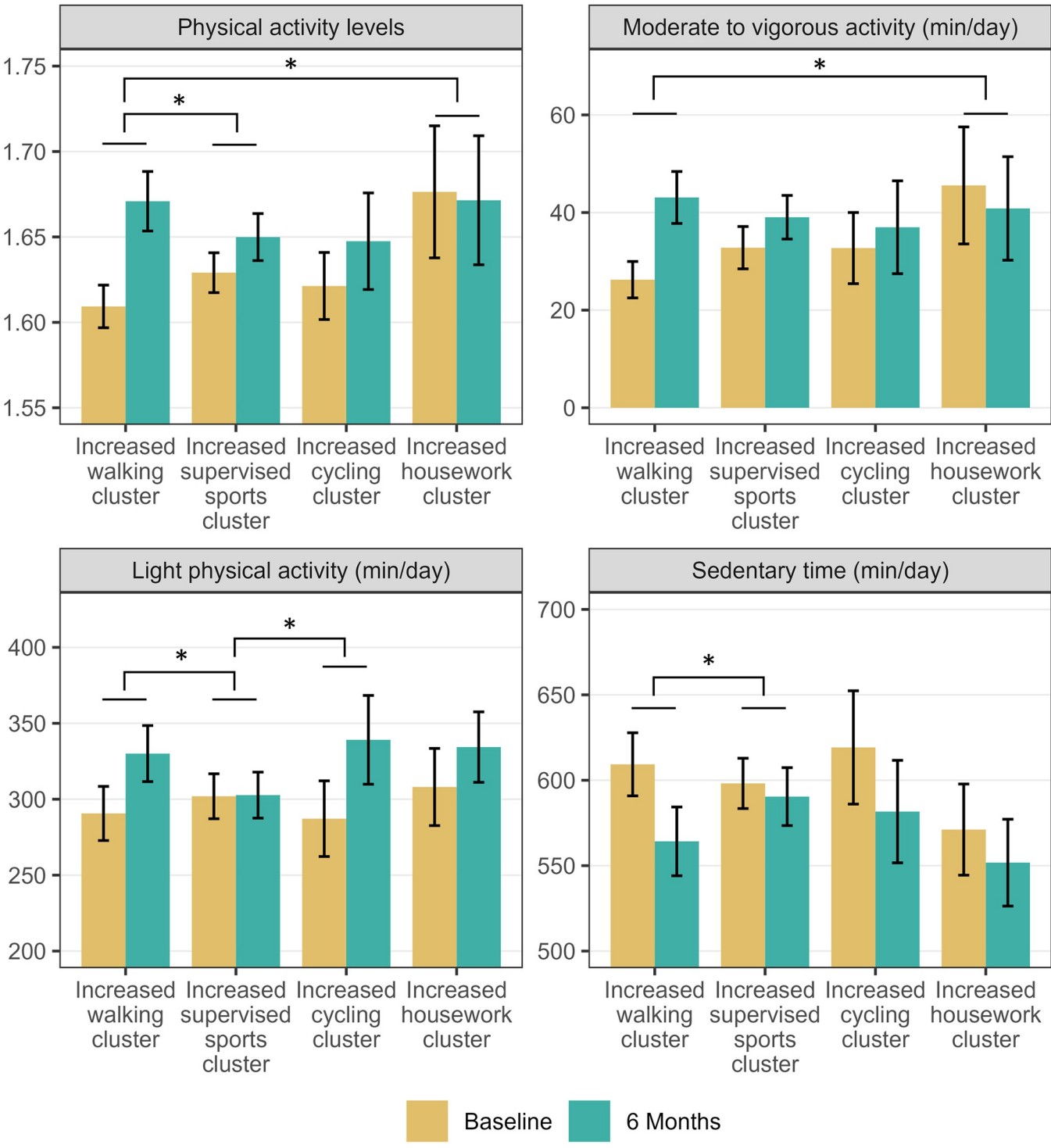

**Fig 2. Differences in accelerometer-measured physical activity and sedentary time changes between the 6 months change clusters.** Data shown as mean ± confidence intervals. Brackets with an asterisk (*) indicate significant differences in changes between clusters.

The change clusters for 12 months did not differ in PAL ($F(2, 225) = 2.655$, p = 0.073, $\eta^2$ = 0.021), nor between SED, LPA and MVPA ($F(6, 444) = 1.131$, p = 0.343; Fig 3). However, the "increased walking and cycling" cluster recorded a significant increase in PAL ($t(85) = 5.02$,

**Table 4. Clusters of how time spent in self-reported physical activity behaviours (minutes per day) changed from baseline to 12 months.**

|  | No change cluster (n = 117) | Increased walking & cycling cluster (n = 86) | Increased supervised sports cluster (n = 29) | Total (n = 232) | Difference between clusters (p-value)[1] |
|---|---|---|---|---|---|
| Walking | -1.1 (37.3) | 20.3 (44.1) | -31.3 (63.7) | 3.1 (46.6) | < 0.001 |
| Cycling | -0.5 (14.7) | 6.4 (12.5) | -0.4 (18.6) | 2.1 (14.8) | 0.003 |
| Unsupervised sports | 2.0 (8.4) | 1.4 (9.1) | -7.6 (21.6) | 0.6 (11.5) | < 0.001 |
| Supervised sports | 1.0 (11.1) | 0.9 (9.2) | 47.5 (38.9) | 6.8 (22.7) | < 0.001 |
| Housework | -0.8 (80.1) | 18.7 (56.3) | 21.3 (96.8) | 9.2 (75.0) | 0.121 |
| Occupational PA | 0.0 (56.2) | 0.2 (37.5) | 2.0 (32.9) | 0.3 (47.3) | 0.979 |
| Gardening | -3.2 (27.4) | 1.8 (19.6) | -2.1 (22.1) | -1.2 (24.1) | 0.340 |
| Sitting | 43.2 (83.5) | -148.0 (101.4) | 2.0 (111.9) | -32.8 (129.7) | < 0.001 |

All measures reported in mean (SD) min•day$^{-1}$. Colours of the cells based on z-transformed values used as cluster inputs (positive–green, 0 –white, negative–red, higher saturation indicates a higher absolute z-value). Activities with highest z-scores of each cluster are bold. PA: physical activity.

[1] p-values of linear model ANOVAs between clusters.

$p < 0.001$) and MVPA ($t(85) = 5.17$, $p < 0.001$) and decreased in SED ($t(85) = -2.68$, $p = 0.009$) from baseline to 12 months whereas the other clusters did not report any significant changes (supplemental material, S13 and S14 Tables).

## Discussion

This study used cluster analysis to identify different domain clusters and clusters of change during the first 12-months of a three-year lifestyle intervention and compared these clusters with device-based measures of PA and SED and estimates of PAL. At baseline, four clusters emerged, characterised by high amounts of cycling and unsupervised sports, walking and housework, low activity and high sitting time, and supervised sports. The "walking and housework" cluster was associated with the highest device-assessed PAL and MVPA, whereas the "cycling" and "inactive" clusters accumulated the most device-assessed LPA and SED, respectively. At six months, four change clusters emerged. The "increased walking" cluster achieved significant improvements in SED, LPA, MVPA and PAL and the "increased cycling" cluster reported the highest increase in LPA. At twelve months, three different clusters, including a larger "no change" cluster, were found with no differences in any variables of movement behaviour.

At baseline, the highest accelerometer-assessed PA, PAL, and lowest SED were found in the "walking and housework" cluster. This finding concurs with previous studies that combined self-reported and device-assessed PA. Specifically, Colley et al. [15] and Scheers et al. [30] found that leisure and transport activities correlate with device-assessed measures of MVPA, whereas occupational and household activities were associated with LPA. However, in the present study, the diary and accelerometer data did not align among other clusters. For example, while cycling is often classified as MVPA [31], the cycling group predominantly recorded LPA, though it is pertinent to note that hip-mounted accelerometers are known to underestimate PA intensity during cycling [32]. Therefore, it is likely that PA intensity was underestimated in this cluster.

No differences were found in the accelerometer-derived variables for the "supervised sports" cluster in comparison to the "inactive" cluster. These findings could be attributed to the relatively short duration of sport activities (20 minutes difference between clusters), an underestimation of intensity in activities such as yoga or weight training, or indeed low associated intensity.

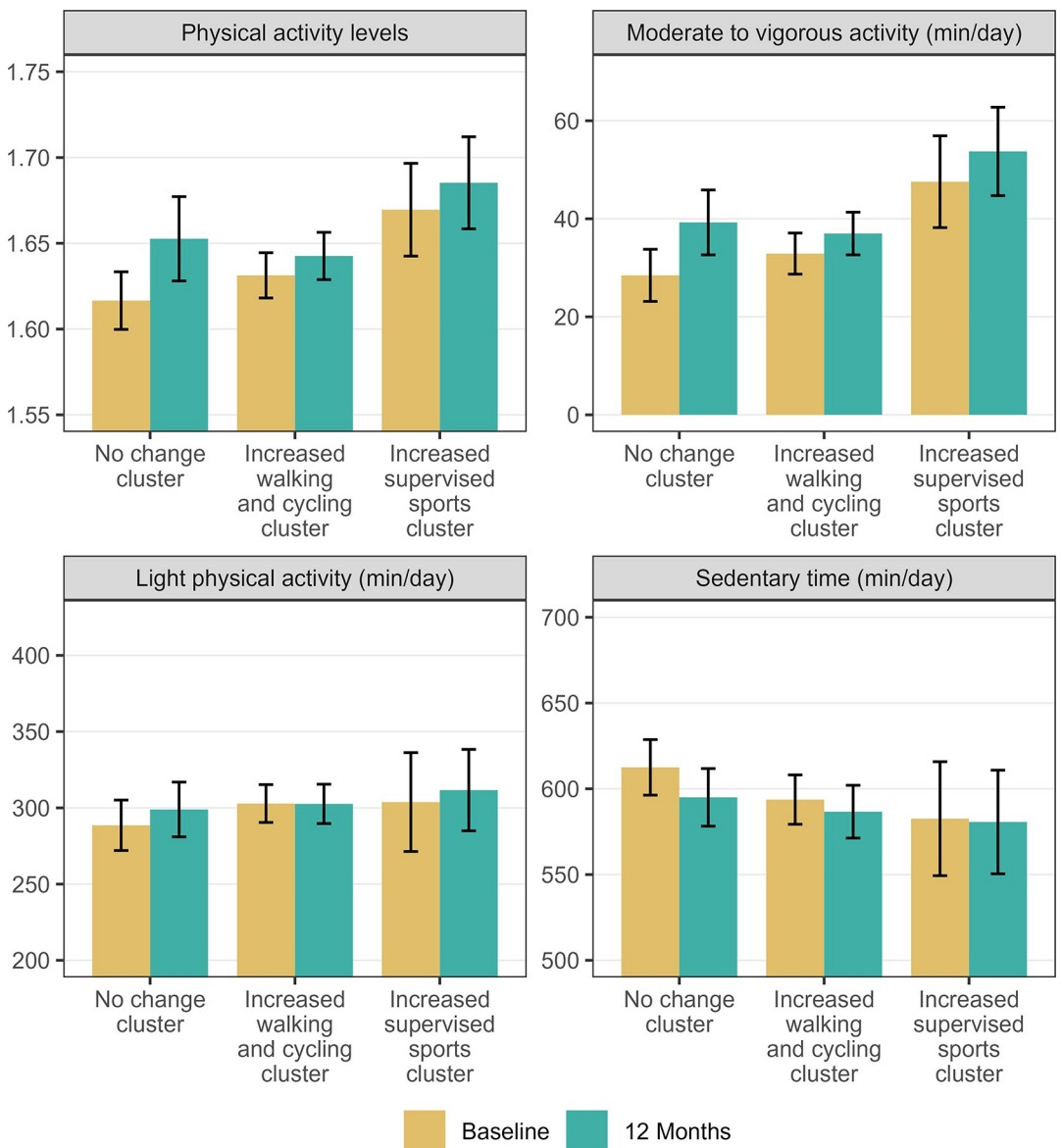

**Fig 3. Differences in accelerometer-measured physical activity and sedentary time changes between the 12 months change clusters.** Data shown as mean ± confidence intervals. There were no significant differences in change between clusters.

In accordance with previous studies [33], self-reported sitting time was much lower than the device-assessed SED, especially for the most active clusters. Indeed, given that research has shown that more active participants underestimate their sitting time [34], it could be postulated that more active individuals have a greater interest in reporting their PA, which is more distinct and easier to recall than periods of sitting. Furthermore, it is pertinent to note that hip-mounted accelerometers assess SED, thereby reflecting a lack of movement, rather than sedentary behaviour, given that it cannot discriminate between postures. Therefore, estimates of SED may encompass lying and sitting but also standing still. Moreover, only leisure and occupational sitting time were recorded in the diary, thereby not considering other contextual factors, such as sitting time accumulated during transportation [35]. The disparity between the self-report and device-assessed SED supports suggestions that these methods should not be

used interchangeably or for purposes of validation [33]. Rather, they can be used to complement one another.

The context in which SED occurs appears to be important. For example, prolonged sitting appears to be more detrimental for metabolic health than the same total amount of sitting accumulated over a long time period with intermittent breaks [36]. Finally, not all types of sitting are associated with risk; while TV viewing is consistently associated with poor health outcomes, occupational sitting is not [37, 38]. Therefore, reporting weekly or average daily volumes alone may not be sufficient.

At six months, four different clusters emerged indicating divergent impact of the intervention programme and approaches to being physically active. Overall, there was no shift towards unsupervised sports, only a small increase in supervised sports. However, two of the four clusters are characterised by a substantial increase in walking and cycling, respectively. In addition, the "increased walking" and "increased cycling" clusters also increased other active leisure and non-leisure behaviours. Subsequently, both were successful in increasing their device-assessed PA, indicating that both walking and cycling were effective for increasing PA and did not necessarily replace other active behaviours.

In contrast, the increase in supervised sports was accompanied by an increase in sitting time and a decline in other active leisure and non-leisure behaviours. Consequently, the smallest decline in device-assessed SED and increases in LPA were recorded in the "increased supervised sports" cluster. This finding may reflect reciprocal compensation between structured exercise and habitual PA. Indeed, some studies have shown that structured exercise, particularly of vigorous-intensity or high volume, prompts a decline in habitual PA due to fatigue and discomfort [39]. This effect is not observed in longer studies (i.e. > six months) [39], suggesting adaptations and increased fitness through prolonged engagement in exercise. Although only a small increase in MVPA was reported in the present study (6 minutes daily), a proportional increase in vigorous intensity PA may have led to a decline in other forms of PA.

Due to their social nature, supervised, e.g., club or team-based sports have been associated with greater improvements in psychosocial health, in addition to improvements attributable to participation in PA, when compared to individual activities [40]. Furthermore, through social connectedness, group-based activities have also been proposed to improve adherence to exercise programmes [41]. Social support is an important component of PA promotion. Indeed, creating social environments that enable change or social networks that reinforce PA behaviour, can effectively increase PA [42]. In the present study, participants did not receive structured exercise prescription, such as exercise classes, but were encouraged to seek support from family and friends. Therefore, social support may have been received through verbal encouragement or indeed by exercising with others. Nonetheless, in the present study the "increased supervised sports" cluster showed the least favourable change in device-based measures of LPA and SED which may have occurred by offsetting increases in supervised sports with decreases in other activities.

The "increased housework" cluster reported a decline in leisure and occupational PA. Despite having less leisure time to be active, this group accumulated a high level of device-assessed PA. However, this group reported a decline in MVPA and PAL which could be attributed to high MVPA at baseline. Congruent with the study of Smith et al. [43], this finding highlights the contribution housework makes to PA, particularly in adults with sedentary jobs or those who are retired. Additionally, non-leisure time activities, such as gardening and occupational PA, may have contributed to higher PA levels. For example, people who stand, walk, lift or carry during occupational and domestic activities have been shown to have higher levels of device-assessed LPA and MVPA and lower SED than those who are predominantly seated during occupational and domestic activities [44].

From baseline to twelve months, three change clusters emerged, with the largest cluster, (50.4% of the participants) showing no change in mode of activities, and therefore no increase in PA behaviours. In PREVIEW, the frequency of group sessions gradually declined from 10 visits in first six months (preparation and action phase) to three in the six months that followed (beginning of the 2.5-y maintenance phase). Frequency of contact is associated with increased effectiveness of PA interventions [45]. Therefore, the decline in contact may explain the large proportion of participants failing to maintain changes in their PA. However, the "increased walking and cycling" cluster did replace sitting time with walking, cycling and housework. These self-report changes corresponded with significant improvements in device-assessed MVPA, SED and PAL and suggest increasing walking, cycling and non-leisure PA are viable steps to improve total PA in pre-diabetic adults.

Sports clubs have been described as an ideal setting to promote PA. Subsequently, many programmes exist that promote PA through sport [46]. However, sport is not for everyone, a lack of confidence, and competence in core skills, negative social comparisons, and negative emotions such as shame are barriers to participation in group sports, specifically in those who were previously inactive [47, 48]. In the present study, the "increased supervised sports" cluster did not increase their device-assessed PA but were the most active cluster at baseline and 12 months, suggesting that supervised sports were preferred by those with pre-existing activity. In contrast, the "increased walking and cycling" cluster were the least active at baseline and the only cluster to report significant improvements in device-based measures at 12 months. In combination, this may suggest that walking and cycling are more accessible ways to increase PA for those who are inactive while supervised sports offer an approach to maintain PA once active.

To our knowledge, this is the first study to combine diary and accelerometery measures to provide contextual information surrounding PA over the first 12 months of a lifestyle intervention [21]. This approach enhances the respective strengths of self-reported and device-assessed PA to gain a greater understanding on PA behaviours over time, the context of which is imperative for future intervention strategies and design. Our study is not without limitations. It is important to acknowledge that the weight loss requirement for participation in the PREVIEW intervention might have led to a more selective sample which might initially be more receptive to lifestyle changes. Furthermore, the analysis was conducted on a selective sample within the PREVIEW intervention who had complete PA diaries. Self-monitoring is an important component of behaviour change interventions and is associated with more successful outcomes in weight loss interventions [49, 50]. Therefore, this sample was likely more motivated and more active than the wider PREVIEW sample (supplementary material; S1 Table). Due to the small, selective sample, some of the clusters contained less than 30 participants, which may limit the reliability and generalisability of those results. Further, stratified analysis, such as by sex or country, were therefore not feasible, thereby precluding additional context and should be considered in future research.

Nevertheless, our analysis emphasises the greater insight gained from blended methodological approaches and provides insight into PA behaviour change during a lifestyle intervention.

Future studies should consider the use of both self-reported and device-based PA measures to provide greater insight into physical activity behaviour and its change. Moreover, smaller recall windows coupled with the use of advancements in ambulatory assessment technology and ecological momentary assessment tools [51, 52], may enhance the recording, and indeed accuracy, of behaviours that change of the course of an intervention.

## Conclusion

This study demonstrated that combining accelerometery and diary data can provide a comprehensive overview of PA behaviour change during a lifestyle intervention. The identification of

PA behaviour clusters may facilitate new, tailored and more sustainable ways to promote PA in different populations. In this intervention, easily accessible activities, such as walking and cycling, were associated with higher PA of all intensities. In contrast, increasing supervised sports was related to a reduction in other activities and was the least effective approach at six months. However, in this selective sample, over half of the participants showed little to no change in their behaviour, highlighting the need for more tailored interventions, especially in the maintenance phase. Further, longitudinal and intervention studies should combine device-based and self-reported PA measures to enhance contextual information and evaluation.

## Supporting information

**S1 Fig. Choice of activities in the physical activity diary.**
(TIF)

**S2 Fig. Survey of sedentary time in the physical activity diary.**
(TIF)

**S1 Table. Comparison of included and excluded participants in the longitudinal PA diary sample.** [1] Linear Model ANOVA, [2] Pearson's Chi-squared test, PA—physical activity, BMI–Body Mass Index, BL—Baseline, PAL–physical activity level; SED–sedentary time, LPA–light physical activity; MVPA–moderate-to-vigorous physical activity.
(DOCX)

**S2 Table. Distribution of age, gender, city, and intervention group between the baseline clusters.** [1] Linear Model ANOVA; [2] Pearson's Chi-squared test.
(DOCX)

**S3 Table. Diary activity z-scores for the baseline clusters.** [1] Linear Model ANOVA, PA–physical activity.
(DOCX)

**S4 Table. Distribution of unsupervised and supervised sports between clusters differentiated by type of activity for the baseline clusters in minutes·day$^{-1}$.**
(DOCX)

**S5 Table. Accelerometer-assessed physical activity and sedentary time in the baseline clusters.** SED, LPA & MVPA are reported in mean (SD) minutes per day. Bold indicates significant differences in the post-hoc analyses between groups with the same superscript letters. PAL: physical activity levels, SED: sedentary time, LPA: light physical activity, MVPA: moderate-to-vigorous physical activity.
(DOCX)

**S6 Table. Distribution of age, gender, city, and intervention group between the baseline to 6 months change clusters.** [1] Linear Model ANOVA; [2] Pearson's Chi-squared test.
(DOCX)

**S7 Table. Diary activity change z-scores for the baseline to 6 months change clusters.** [1] Linear Model ANOVA, PA–physical activity
(DOCX)

**S8 Table. Distribution of unsupervised and supervised sports change between clusters differentiated by type of activity for the baseline to 6 months change clusters in minutes·day$^{-1}$.**
(DOCX)

**S9 Table. Accelerometer-assessed physical activity and sedentary time at baseline and their change after six months for the six-month change clusters.** SED, LPA & MVPA are reported in mean (SD) minutes per day. Bold indicates significant differences in the *post-hoc* analyses between groups with the same superscript letters. PAL: physical activity levels, SED: sedentary time, LPA: light physical activity, MVPA: moderate-to-vigorous physical activity, PA: physical activity.
(DOCX)

**S10 Table. Distribution of age, gender, city, and intervention group between the baseline to 12 months change clusters.** [1] Linear Model ANOVA; [2] Pearson's Chi-squared test.
(DOCX)

**S11 Table. Diary activity change z-scores for the baseline to 12 months change clusters.** [1] Linear Model ANOVA, PA–physical activity.
(DOCX)

**S12 Table. Distribution of unsupervised and supervised sports change between clusters differentiated by type of activity for the baseline to 12 months change clusters in minutes·day$^{-1}$.**
(DOCX)

**S13 Table. Accelerometer-assessed physical activity and sedentary time at baseline and their change after 12 months for the 12-month change clusters.** SED, LPA & MVPA are reported in mean (SD) minutes per day. PAL: physical activity levels, SED: sedentary time, LPA: light physical activity, MVPA: moderate-to-vigorous physical activity, PA: physical activity.
(DOCX)

**S14 Table. Change of device-based PA from baseline to 12 months for the change clusters (Results of individual one-sample t-tests).** Bonferroni-adjusted $\alpha = 0.0167$ to account for multiple testing; significant results are bold.
(DOCX)

## Acknowledgments

We are especially grateful to the participants for volunteering their time to the study and acknowledge the research staff from each site. From UCPH: Laura Pastor-Sanz, Grith Møller, Lone Vestergaard Nielsen, Arne Astrup, Finn Sandø-Pedersen, Morten Bo Johansen, Ulla Skovbæch Pedersen, Maria Roed Andersen, Marianne Juhl Hansen, Jane Jørgensen, Sofie Skov Frost, Lene Stevner. From HEL: Saara Kettunen, Tiia Kunnas, Sanna Ritola, Laura Korpipää, Heini Hyvärinen, Karoliina Himanen, Tiina Pellinen, Elina Malkamäki, Heidi Jokinen, Pauliina Kokkonen, Liisi Korhonen, Jaana Valkeapää, Heli Pikkarainen, Martta Nieminen, Tuulia Ingman, Pihla Mäkinen, and Sonja Toijonen. From UNOTT: Clare Randall, Nicky Gilbert, Shelley Archer, Sally Maitland, Melanie Marshall, Cheryl Percival, Jakki Pritchard, and Laura Helm. From UNAV: Blanca Martinez de Morentin, Maria Hernandez Ruiz de Eguilaz, Salome Perez Diez, Veronica Ciaurriz, Angels Batlle, Maria Jose Cobo. From MU: Pavlina Gateva, Rossica Metodieva, Galia Dobrevska. From SU: Kelly A Mackintosh, Melitta A. McNarry, Jeff Stephens, Gareth Dunseath, Steve Luzio, Masoumeh Minou. From THL: Merja Tukiainen, Ira Greinert, Laura Karjalainen, Jukka Lauronen. From UNSYD: Fiona Atkinson, Michele Whittle, Jessica Burke, Kylie Simpson, Kimberley Way, Sally McClintock, Radhika Seimon, Shelly Keating, Kirsten Bell, Tania Markovic, Cathy Corry, Evalyn Eldering, Ian

Caterson. From UOA: Jim Mann (University of Otago), Boyd Swinburn, Lindsay Plank, Nicholas Gant, Jon Woodhead, Anne-Thea McGill, Katya Volkova, Madhavi Bollineni, Clarence Vivar, Kelly Storey, Niamh Brennan, Audrey Tay.

## Author Contributions

**Conceptualization:** Gareth Stratton, Mikael Fogelholm, Ian Macdonald.

**Data curation:** Mathijs Drummen, J. Alfredo Martinez, Santiago Navas-Carretero, Teodora Handjieva-Darlenska, Georgi Bogdanov, Nicholas Gant, Sally D. Poppitt, Marta P. Silvestre, Jennie Brand-Miller, Roslyn Muirhead, Wolfgang Schlicht, Maija Huttunen-Lenz, Shannon Brodie, Elli Jalo, Margriet Westerterp-Plantenga, Tanja Adam, Pia Siig Vestentoft, Heikki Tikkanen, Jonas S. Quist, Nils Swindell.

**Formal analysis:** Leon Klos.

**Funding acquisition:** Anne Raben.

**Investigation:** Mikael Fogelholm, Mathijs Drummen, Ian Macdonald, J. Alfredo Martinez, Santiago Navas-Carretero, Teodora Handjieva-Darlenska, Georgi Bogdanov, Nicholas Gant, Sally D. Poppitt, Marta P. Silvestre, Jennie Brand-Miller, Roslyn Muirhead, Wolfgang Schlicht, Maija Huttunen-Lenz, Shannon Brodie, Elli Jalo, Margriet Westerterp-Plantenga, Tanja Adam, Pia Siig Vestentoft, Heikki Tikkanen, Jonas S. Quist, Anne Raben.

**Methodology:** Gareth Stratton, Kelly A. Mackintosh, Melitta A. McNarry, Ian Macdonald, J. Alfredo Martinez, Santiago Navas-Carretero, Teodora Handjieva-Darlenska, Georgi Bogdanov, Jennie Brand-Miller, Roslyn Muirhead, Wolfgang Schlicht, Maija Huttunen-Lenz, Elli Jalo, Margriet Westerterp-Plantenga, Pia Siig Vestentoft, Jonas S. Quist, Anne Raben.

**Project administration:** Anne Raben.

**Supervision:** Gareth Stratton, Nils Swindell.

**Writing – original draft:** Leon Klos.

**Writing – review & editing:** Gareth Stratton, Kelly A. Mackintosh, Melitta A. McNarry, Mikael Fogelholm, J. Alfredo Martinez, Sally D. Poppitt, Marta P. Silvestre, Jennie Brand-Miller, Roslyn Muirhead, Maija Huttunen-Lenz, Shannon Brodie, Elli Jalo, Margriet Westerterp-Plantenga, Tanja Adam, Pia Siig Vestentoft, Heikki Tikkanen, Jonas S. Quist, Anne Raben, Nils Swindell.

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
