## [Decision Letter · Decision Letter 0]

2 Jan 2024

PONE-D-23-36535Combining diaries and accelerometers to explain change in physical activity during a lifestyle intervention for adults with pre-diabetes: a PREVIEW sub-studyPLOS ONE

Dear Dr. Swindell,

Thank you for submitting your manuscript to PLOS ONE. After careful consideration, we feel that it has merit but does not fully meet PLOS ONE’s publication criteria as it currently stands. Therefore, we invite you to submit a revised version of the manuscript that addresses the points raised during the review process.

We look forward to receiving your revised manuscript.

Kind regards,

Zulkarnain Jaafar

Academic Editor

PLOS ONE

Journal Requirements:

"The PREVIEW study received grants from the EU 7th Framework Programme (FP7-KBBE-2012), grant no: 312057; the New Zealand Health Research Council, grant no. 14/191; and the NHMRC-EU Collaborative Grant, Australia."

"Ian MacDonald: Principle investigator of the Nottingham site, conception of the study, editing of the manuscript.

International Life Sciences Institute (ILSI) Europe: Member of Dietary Carbohydrates Task Force, Member of expert group on 'Efficacy Markers of Diabetes Risk'. Travel and subsistence paid but no attendance fees. 

Nature Publishing Group (Springer Nature) - Editor International Journal of Obesity.  Travel and accommodation re-imbursed and honorarium paid (Amount received per annum over £5,000: yes). 

Nestle Research - Scientific Advisory Board. Travel and accommodation reimbursed, and honorarium paid (Amount received per annum over £5,000: yes)

Mars Incorporated-Waltham Centre for Pet Nutrition Peer-review of pet nutrition research projects honorarium received (Amount received per annum over £5,000: no); 

Mars UK/Europe - Member of Nutrition Advisory Board, and Health and Wellbeing Committee travel and subsistence costs reimbursed. Honorarium paid to the University of Nottingham.

Mars Incorporated - Member of Mars Scientific Advisory Committee. Honorarium paid to the University of Nottingham.

Novozymes - Member of Science Advisory Board. Travel and subsistence costs reimbursed. Honorarium paid to the University of Nottingham. 

Public Health England - Member of Scientific Advisory Committee on Nutrition. Travel and accommodation re-imbursed and honorarium paid (Amount received per annum over £5,000: no)

Jennie Brand-Miller: Principle investigator of the Sydney site and participated in editing of the manuscript. Jennie Brand-Miller is President and Director of the Glycemic Index Foundation and oversees a glycemic index testing service at the University of Sydney. She receives royalties for books about carbohydrates, obesity and diabetes."

5. We note you have included a table to which you do not refer in the text of your manuscript. Please ensure that you refer to Table 4,6 and 7 in your text; if accepted, production will need this reference to link the reader to the Table.

Additional Editor Comments:

**ACADEMIC EDITOR: **Dear Author,Please attend to all the comments provided by the reviewers. Please ensure comprehensive correction of the statistical aspects. The decision of this manuscript is justified based on PLOS ONE’s publication criteria and not on its novelty or perceived impact.

Reviewers' comments:

Reviewer's Responses to Questions

**Comments to the Author**

1. Is the manuscript technically sound, and do the data support the conclusions?

Reviewer #1: Yes

Reviewer #2: Yes

Reviewer #3: Yes

2. Has the statistical analysis been performed appropriately and rigorously? 

Reviewer #1: Yes

Reviewer #2: Yes

Reviewer #3: Yes

3. Have the authors made all data underlying the findings in their manuscript fully available?

Reviewer #1: No

Reviewer #2: No

Reviewer #3: Yes

4. Is the manuscript presented in an intelligible fashion and written in standard English?

Reviewer #1: Yes

Reviewer #2: Yes

Reviewer #3: Yes

5. Review Comments to the Author

Reviewer #1: The authors intended to leverage self-report and device-based physical activity measures to identify domain clusters and behavior change during an intervention for adults with prediabetes.

1. Only people with an initial weight loss of over 8% were included. However, the abstract did not include such inclusion/exclusion criteria. How will these criteria affect the conclusions and their implications for the result interpretation?

2. PREVIEW has two diet and two PA arms. Please make it more specific. Also, how were the subjects in the current study sampled, and how generalizable were the results obtained from such an intervention study?

3. The accelerometer cannot record water-based activities. Will this be a concern or bias in data collection? How many percent of participants have water-based activities, e.g., swimming?

4. Please clarify how the Z-scores of the activities were calculated.

5. The exclusion proportion for each country varies significantly. However, the country was not adjusted in the analysis. What are the implications of current data?

Reviewer #2: This was a well written and well performed paper on the topic of combining data from accelerometers with questionnaire data. Following are my comments on the paper:

1. It was a little unclear when the accelerometer data was collected, I assumed it was at the same time points as the diary data but this was not stated

2. It would be helpful to write out the cut points that you used, as these can differ quite a lot

3. The second sentence under the heading of “diary data” is quite confusing. It sounds like you are saying that winter sports can be an occupational activity and then you mention occupational activity again.

4. Did the participants write just the duration of each activity or also the time point? Was the time matched to the accelerometer data i.e. if I wrote that I went jogging from 10:00-11:20 was that matched with the accelerometer data? I was a little confused to understand what you meant with the PA diary entries and accelerometer wear-time were aligned at each time-point (line 185).

5. There was a huge drop out which I feel needs to be discussed more. Only 10% of your sample fulfilled the criteria regarding acc data and diary data at each time point. This seems really low and it could be some unknown factor that separates these people from the other participants.

6. I also wondered why they needed to have complete data for all three timepoints. Since you never compared all three at the same time shouldn’t it be enough to have timepoint 0 and 1, and timepoint 0 and 2 respectively?

7. I saw in the supplementary table that those that were included were quite different on acc measured physical activity so this needs to be discussed in the paper and mentioned in the results.

8. Consider whether Tables 3, 5 and 7 might be better represented in a figure. You could have brackets with stars to indicate comparisons that were significant.

9. I found the order of the clusters a little confusing. It would be best to have the incactivity cluster first because that is what feels best to compare the other clusters with, the same with the no change cluster.

10. Since your sample is quite small some of the groups are very small. There needs to be a discussion on the robustness and reliability of the results when the final sample is so small.

Reviewer #3: The authors in this article proposed the combination of self-report and device-based measures which seems to be a good solution, but It remains an open question whether it is better than other combined forms of measurement? Each activity measurement method has its limitations

It is clear that combining the two control methods provides greater insight into behaviors that change during the intervention.

We are still looking for universal measurement methods, therefore the attempt undertaken by the authors is of undoubted value for further research.The manuscript describes a technically sound piece of scientific research, including data supporting the conclusions. The experiments were performed correctly, with appropriate control, replication, and sample sizes. Conclusions are drawn correctly based on the presented data.

6. PLOS authors have the option to publish the peer review history of their article (what does this mean?). If published, this will include your full peer review and any attached files.

Reviewer #1: No

Reviewer #2: No

Reviewer #3: No

---

## [Author Response · Author response to Decision Letter 0]

20 Feb 2024

We would like to thank the reviewers for their thoughtful comments and helpful feedback, which we believe has improved our manuscript. We have addressed each point in turn below, cross-referencing with line numbers, and using tracked changes in the manuscript to easily identify amendments. 

Reviewer #1: The authors intended to leverage self-report and device-based physical activity measures to identify domain clusters and behaviour change during an intervention for adults with prediabetes.

1. Only people with an initial weight loss of over 8% were included. However, the abstract did not include such inclusion/exclusion criteria. How will these criteria affect the conclusions and their implications for the result interpretation?

We did not include this requirement in the abstract to keep it within the word limit. The low-calorie diet before the intervention was undertaken to give the participants a sense of achievement. People who did not achieve the weight loss goal might be more likely to not adhere to the intervention instructions. On the other hand, people included in the trial might have a higher motivation at the beginning of the intervention given the recently achieved weight loss. To conclude, the weight loss requirement might have led to a more selective sample which might initially be more receptive to lifestyle changes. To clarify thnis point, a short sentence has been added to the limitation section [lines 437-475]. 

2. PREVIEW has two diet and two PA arms. Please make it more specific. Also, how were the subjects in the current study sampled, and how generalizable were the results obtained from such an intervention study?

We have added a brief description of the diet and exercise interventions and randomisation [lines 148-150] and highlighted the primary methods paper in which more detail can be found. Participants were recruited using multiple methods across the eight study sites ( e.g., newspaper advertisements, newsletters, radio and television advertisements/interviews) and direct contact with primary and occupational health care providers, which is stated in the study protocol. 

We want to emphasize that we only used a small sub-sample of the study for our analysis and have ensured this is clearly stated in the limitations section of the discussion. Our main focus was not the effect of the different interventions but the methodological approach of combining physical activity measurement methods, and therefore we believe the sub-sample used and approach taken are appropriate. 

3. The accelerometer cannot record water-based activities. Will this be a concern or bias in data collection? How many percent of participants have water-based activities, e.g., swimming?

We agree this is an impotant consideration. Indeed, 23.1%-26.1% of participants stated they engaged in water-based activities, and of these, there was an average 12 minutes of water-based activities per day. Moreover, S4 Table shows that water-based activities account for up to 3.5 minutes per day with differences of less than two minutes between clusters. Therefore, it is unlikely that removing the accelerometer for water-based activities will have had a large impact on, or introduced significant bias to, the comparison of accelerometer-based physical activity between clusters. 

4. Please clarify how the Z-scores of the activities were calculated.

Apologies for this lack of clarity. Z-Scores at baseline were calculated for each physical activity behaviour (as used for the cluster analysis, such as walking, cycling, or unsupervised sports) separately. The mean value for a particular behaviour was subtracted from each individual value, with the result then divided by the standard deviation. For the change data, the z-scores were calculated for the change of time spent within each physical activity behaviour. This has now been clarified in the methods section [lines 210-211].

5. The exclusion proportion for each country varies significantly. However, the country was not adjusted in the analysis. What are the implications of current data?

The majority of included participants were from northern and western European countries, which we acknowledge is a limitation. Small numbers of participants from some countries precluded country-specific differences being investigated, which has now been incorporated as a limitation to the discussion [lines 473-476]. Whilst the distribution of activities might differ significantly between countries, the comparison of diary and accelerometer data would unlikely be influenced by country.

Reviewer #2: This was a well written and well performed paper on the topic of combining data from accelerometers with questionnaire data. Following are my comments on the paper:

1. It was a little unclear when the accelerometer data was collected, I assumed it was at the same time points as the diary data but this was not stated

Participants wore an accelerometer and completed the diary a week prior to the clinical investigation days at baseline, 6 and 12 months [lines 162-163]. For each participant and measurement point, there are a maximum of seven days with overlapping accelerometer and diary data. Participants were included only if there were at least four days of valid accelerometer data and valid diary data (of the same day) at each time-point [line 192]. 

2. It would be helpful to write out the cut points that you used, as these can differ quite a lot

Thank you. We have added the cut-points to the manuscript [lines 174-175].

3. The second sentence under the heading of “diary data” is quite confusing. It sounds like you are saying that winter sports can be an occupational activity and then you mention occupational activity again.

Thank you for highlighting this lack of clarity. We have rephrased this sentence [lines 179-180] to “The diary consisted of pre-selected activities, such as walking, cycling, winter sports, gymnastics, dancing, fishing and hunting, jogging, team sports, water sports, housework, gardening and occupational activities“. 

4. Did the participants write just the duration of each activity or also the time point? Was the time matched to the accelerometer data i.e. if I wrote that I went jogging from 10:00-11:20 was that matched with the accelerometer data? I was a little confused to understand what you meant with the PA diary entries and accelerometer wear-time were aligned at each time-point (line 185).

As can be seen in Figures 1 and 2 in the supplement, the diaries only asked for the duration, perceived level of exertion, and heart rate. Therefore, we could only match the data at day level rather than to individual activities during the day. To clarify, we have rephrased the sentence [lines 187-188]. 

5. There was a huge drop out which I feel needs to be discussed more. Only 10% of your sample fulfilled the criteria regarding acc data and diary data at each time point. This seems really low and it could be some unknown factor that separates these people from the other participants.

As can be seen in Tables 1 and S1 (supplement), there are differences between the participants that were included or excluded, for example in terms of age, BMI and accelerometer-measured PA. As shown in Table S1, only 810 participants had valid accelerometer data at the 12-month time-point. Although not included in this table, diary entries also decreased significantly at 6 and 12 months, thereby leaving this small and selective sample. We have now clarified this in the discussion [lines 468-473]. 

6. I also wondered why they needed to have complete data for all three timepoints. Since you never compared all three at the same time shouldn’t it be enough to have timepoint 0 and 1, and timepoint 0 and 2 respectively?

We wanted to limit the analysis to one sample. Our initial idea was to also look at the changes between 6 and 12 months, however, we decided against this step given the length of the manuscript and the lack of additional novel findings. Still, only having one sample, we could conclude that, although participants’ PA patterns changed at 6 months, these changes disappeared when looking at a longer time span of 12 months. By including more participants in the 6 months change analysis we would have introduced some bias when comparing the 6 and 12 months change results. 

7. I saw in the supplementary table that those that were included were quite different on acc measured physical activity so this needs to be discussed in the paper and mentioned in the results.

Thank you for highlighting this. We have now incorporated a sentence in the results to highlight that MVPA was higher in our selective sample [lines 233-234], as well as acknowledging this in the discussion [line 472].

8. Consider whether Tables 3, 5 and 7 might be better represented in a figure. You could have brackets with stars to indicate comparisons that were significant.

We agree. We have now provided plots (Figs 1-3) instead of tables to show the differences in device-based measures between clusters in the manuscript, and incorporated the tables in the supplement for further information.

9. I found the order of the clusters a little confusing. It would be best to have the inactivity cluster first because that is what feels best to compare the other clusters with, the same with the no change cluster.

Thank you. We agree and have rearranged the clusters in the tables and figures, with the inactive cluster at baseline (Table 2 & Fig 1) and the no change cluster (baseline – 12 months; Table 4 & Fig 3) now first.

10. Since your sample is quite small some of the groups are very small. There needs to be a discussion on the robustness and reliability of the results when the final sample is so small.

Thank you for this comment, we agree and have added the following sentence to the limitations section [lines 473-476] “Due to the small, selective sample, some of the clusters contained less than 30 participants, which may limit the reliability and generalisability of those results. Further, stratified analysis,such as by sex or country, were therefore not feasible, thereby precluding additional context and should be considered in future research“.

Reviewer #3: The authors in this article proposed the combination of self-report and device-based measures which seems to be a good solution, but It remains an open question whether it is better than other combined forms of measurement? Each activity measurement method has its limitations

It is clear that combining the two control methods provides greater insight into behaviors that change during the intervention.

We are still looking for universal measurement methods, therefore the attempt undertaken by the authors is of undoubted value for further research.The manuscript describes a technically sound piece of scientific research, including data supporting the conclusions. The experiments were performed correctly, with appropriate control, replication, and sample sizes. Conclusions are drawn correctly based on the presented data.

Thank you for your comment.

---

## [Decision Letter · Decision Letter 1]

4 Mar 2024

Combining diaries and accelerometers to explain change in physical activity during a lifestyle intervention for adults with pre-diabetes: a PREVIEW sub-study

PONE-D-23-36535R1

Dear Dr. Swindell,

We’re pleased to inform you that your manuscript has been judged scientifically suitable for publication and will be formally accepted for publication once it meets all outstanding technical requirements.

Kind regards,

Zulkarnain Jaafar

Academic Editor

PLOS ONE

Additional Editor Comments (optional):

Reviewers' comments:

Reviewer's Responses to Questions

**Comments to the Author**

1. If the authors have adequately addressed your comments raised in a previous round of review and you feel that this manuscript is now acceptable for publication, you may indicate that here to bypass the “Comments to the Author” section, enter your conflict of interest statement in the “Confidential to Editor” section, and submit your "Accept" recommendation.

Reviewer #1: All comments have been addressed

Reviewer #2: All comments have been addressed

2. Is the manuscript technically sound, and do the data support the conclusions?

Reviewer #1: Yes

Reviewer #2: (No Response)

3. Has the statistical analysis been performed appropriately and rigorously? 

Reviewer #1: Yes

Reviewer #2: (No Response)

4. Have the authors made all data underlying the findings in their manuscript fully available?

Reviewer #1: Yes

Reviewer #2: (No Response)

5. Is the manuscript presented in an intelligible fashion and written in standard English?

Reviewer #1: Yes

Reviewer #2: (No Response)

6. Review Comments to the Author

Reviewer #1: All raised comments have been successfully addressed. i have no further comments on this manuscript.

Reviewer #2: (No Response)

7. PLOS authors have the option to publish the peer review history of their article (what does this mean?). If published, this will include your full peer review and any attached files.

Reviewer #1: No

Reviewer #2: No

---

## [Editor Report · Acceptance letter]

12 Mar 2024

PONE-D-23-36535R1 

PLOS ONE

Dear Dr. Swindell, 

I'm pleased to inform you that your manuscript has been deemed suitable for publication in PLOS ONE. Congratulations! Your manuscript is now being handed over to our production team.

Kind regards, 

on behalf of

Dr. Zulkarnain Jaafar 

Academic Editor

PLOS ONE